# Proteomic Analysis of Retinal Mitochondria-Associated ER Membranes Identified Novel Proteins of Retinal Degeneration in Long-Term Diabetes

**DOI:** 10.3390/cells11182819

**Published:** 2022-09-09

**Authors:** Joshua J. Wang, Karen Sophia Park, Narayan Dhimal, Shichen Shen, Xixiang Tang, Jun Qu, Sarah X. Zhang

**Affiliations:** 1Department of Ophthalmology, Ross Eye Institute, Jacobs School of Medicine and Biomedical Sciences, University at Buffalo, State University of New York, Buffalo, NY 14203, USA; 2Department of Biochemistry, Jacobs School of Medicine and Biomedical Sciences, University at Buffalo, State University of New York, Buffalo, NY 14203, USA; 3Department of Pharmaceutical Sciences, School of Pharmacy and Pharmaceutical Sciences, University at Buffalo, State University of New York, Buffalo, NY 14214, USA; 4Department of Endocrinology and Metabolism and VIP Medical Service Center, Guangdong Provincial Key Laboratory of Diabetology, The Third Affiliated Hospital, Sun Yat-Sen University, Guangzhou 510630, China

**Keywords:** diabetic retinopathy, Type 1 diabetes, mitochondria-associated ER membrane, proteomic, retinal degeneration

## Abstract

The mitochondria-associated endoplasmic reticulum (ER) membrane (MAM) is the physical contact site between the ER and the mitochondria and plays a vital role in the regulation of calcium signaling, bioenergetics, and inflammation. Disturbances in these processes and dysregulation of the ER and mitochondrial homeostasis contribute to the pathogenesis of diabetic retinopathy (DR). However, few studies have examined the impact of diabetes on the retinal MAM and its implication in DR pathogenesis. In the present study, we investigated the proteomic changes in retinal MAM from Long Evans rats with streptozotocin-induced long-term Type 1 diabetes. Furthermore, we performed in-depth bioinformatic analysis to identify key MAM proteins and pathways that are potentially implicated in retinal inflammation, angiogenesis, and neurodegeneration. A total of 2664 unique proteins were quantified using IonStar proteomics-pipeline in rat retinal MAM, among which 179 proteins showed significant changes in diabetes. Functional annotation revealed that the 179 proteins are involved in important biological processes such as cell survival, inflammatory response, and cellular maintenance, as well as multiple disease-relevant signaling pathways, e.g., integrin signaling, leukocyte extravasation, PPAR, PTEN, and RhoGDI signaling. Our study provides comprehensive information on MAM protein changes in diabetic retinas, which is helpful for understanding the mechanisms of metabolic dysfunction and retinal cell injury in DR.

## 1. Introduction

Diabetic Retinopathy (DR) is a common complication of diabetes and the leading cause of blindness in working-age adults worldwide [1]. While DR had been considered a major microvascular disease that primarily affects the retinal vascular system, compelling evidence from the past two decades suggests that retinal dysfunction and neuronal degeneration are early and important manifestations of DR [2,3,4,5,6]. In some diabetic patients, the thinning of the retinal neural fiber layer was detected prior to visible vascular lesions [7]. Moreover, the retina is a complex neural tissue consisting of over one hundred subtypes of neuron that are organized in layers. The structural deterioration of the neural retina, such as disorganization of the retinal inner layers identified by optical coherence tomography, has been associated with function deficits in electroretinogram [8], loss of contrast sensitivity [6,9], and reduced visual acuity [10,11] in individuals with DR. Furthermore, histological studies have confirmed loss of retinal ganglion cells and neuronal cell death in retinal tissues from diabetic animals and postmortem donor eyes with DR [12]. Collectively, these findings suggest that neurodegeneration, in addition to vasculopathy, contributes to vision impairment in DR. However, the exact mechanisms underlying diabetic neurovascular damage are poorly understood.

The endoplasmic reticulum (ER) and mitochondria are two vital cellular compartments that govern the foremost important biological processes including protein and sterol synthesis, cellular metabolism and energy production, calcium homeostasis, and redox status. The dysfunction of the ER and mitochondria has been observed in retinal cells during diabetes [13,14,15,16,17]. Intriguingly, in many pathological conditions, disturbed ER homeostasis that results in ER stress and calcium dysregulation is often associated with mitochondrial damage, metabolic defects, and oxidative stress [18,19,20]. The close relationship between the two vital organelles has been further confirmed by the identification of the mitochondria-associated ER membrane (MAM), a specialized subdomain of the ER that physically and biochemically connects the ER with the mitochondria [21]. Functionally, the MAM serves as a major platform for calcium metabolism. In normal conditions, a constitutively low level of calcium is released from the ER to maintain the calcium signaling and ATP production in the mitochondria. Disturbances in calcium trafficking from the ER to the mitochondria could lead to defects in mitochondrial respiration, reductions in ATP production, and even cell death [22,23]. In addition, the MAM can regulate reactive oxygen species (ROS) generation. For example, PERK, which has been shown to be activated by ER stress in diabetic retina [14,15,16,17], facilitates the propagation of ROS signals at the MAM and promotes ROS-mediated apoptosis [24]. Thus, understanding the impact of diabetes on MAM in retinal cells could provide new insights into neuronal and vascular pathologies in DR.

In the present study, we isolated retinal MAM from Long Evans rats with streptozotocin (STZ)-induced long-term Type 1 diabetes and investigated changes in the proteome profile of retinal MAM using ion current-based nanoLiquid Chromatography (LC)-Mass Spectrometry (MS) analysis. We conducted bioinformatic analyses to identify differentially expressed MAM proteins in Type 1 diabetes and examine their implications in pathways of retinal inflammation, angiogenesis, and neurodegeneration in DR.

## 2. Materials and Methods

### 2.1. Animals

Long Evans Rats (Strain Code: 006) were purchased from Charles River Laboratories (Raleigh, NC, USA). To induce diabetes, 8-week-old male rats were randomly assigned to receive a single peritoneal injection of streptozotocin (60 mg/kg body weight; Sigma-Aldrich, St. Louis, MO, USA, dissolved in 0.01 mol/l citrate buffer, pH 4.5) or vehicle (0.01 mol/L citrate buffer, pH 4.5) as a control. Blood glucose levels were measured using a glucometer and test strips (ReliOn Prime, Bentonville, AR, USA). Animals with persistent blood glucose levels greater than 250 mg/dL were considered diabetic. At 10 months after the onset of diabetes, the rats were humanely euthanized, and their retinas were harvested for MAM isolation. All animal procedures were approved by the Institutional Animal Care and Use Committees at University at Buffalo, State University of New York, and in accordance with the guidelines of the Association for Research in Vision and Ophthalmology statements for the “Use of Animals in Ophthalmic and Vision Research”.

### 2.2. Isolation of MAM from Rat Retinas

The MAM were isolated following the protocol as previously described [25] with minor modifications (Figure 1). Six retinas from 3 rats were pooled as one sample and manually homogenized on ice. Nuclei and unbroken cells were separated by centrifugation at 740× *g* for 10 min. Further centrifugation of the supernatant at 9000× *g* for 10 min was performed to isolate the crude mitochondria from the pellet. Following three washes, the crude mitochondrial fraction was resuspended in mitochondrial re-suspension buffer (MRB, 250-mM mannitol, 5-mM HEPES, pH 7.4, and 0.5-mM EGTA) to a final volume of 2.0 mL, and layered on top of Percoll medium (225-mM mannitol, 25-mM HEPES, pH 7.4, 1-mM EGTA and 30% Percoll (*v*/*v*)). After centrifugation at 95,000× *g* for 30 min, a dense band containing purified mitochondria was localized approximately at the bottom of the tube, and the crude MAM fraction was visible as the diffused white band located above the mitochondria. The MAM fraction was collected and washed to remove the Percoll by centrifugation at 6300× *g* for 10 min followed by further purified by centrifugation of the supernatant at 100,000× *g*. All the fractions were flash frozen by liquid nitrogen and preserved in −80 °C until use.

### 2.3. MAM Sample Preparation

Collected MAM samples were dissolved in a surfactant cocktail buffer containing 50 mM Tris-formic acid (FA), 150 mM NaCl, 2% sodium dodecyl sulfate (SDS), 1% sodium deoxycholate, and 2% IGEPAL^®^ CA-630. Protease and phosphatase inhibitor tablets (Roche Applied Science, Indianapolis, IN, USA) were also supplemented to the buffer. Samples were first settled on ice for 10 min, and were sonicated for 30 s (non-continuously, 5 s as a burst) using a high-energy probe (Qsonica, Newton, CT, USA) until the liquid becomes pellucid. Samples were again settled on ice for another 30 min and were then centrifuged at 20,000× *g* under 4 °C for 30 min. The supernatant portion was carefully transferred to new Eppendorf tubes, and protein concentrations for individual samples were determined by bicinchoninic acid assay (BCA; Pierce Biotechnology, Inc., Rockford, IL, USA).

For proteolytic digestion, 100 μg of extracted proteins were used for each sample (normalized to 100 μL with 0.5% SDS). Protein extracts were reduced by 10 mM dithiothreitol (DTT) under 56 °C for 30 min, and were alkylated by 20 mM iodoacetamide (IAM) under 37 °C for 30 min. Both steps were performed in an Eppendorf thermomixer with aluminum foil cover and constant vortexing. Proteins were precipitated by the stepwise addition of 7 volumes of chilled acetone, and the mixture was incubated under −20 °C for 3 h. Precipitated proteins were pelleted by centrifugation at 20,000× *g* under 4 °C for 30 min, and the supernatant was discarded. Protein pellets were gently washed with 500 μL methanol, air-dried for 1 min, and saturated by 80 μL 50 mM Tris-FA. A total of 5 μg trypsin (Sigma-Aldrich, St. Louis, MO, USA) at a concentration of 0.25 μg/mL was added at a final enzyme:substrate (E:S) ratio of 1:20 (*w*/*w*), and the mixture was incubated under 37 °C for 6 h. The digestion reaction was terminated by addition of 1 μL FA, and samples were centrifuged (20,000× *g*, 4 °C, 30 min) and transferred to LC vials for analysis.

### 2.4. Liquid Chromatography (LC)-Mass Spectrometry (MS) Analysis

An optimized nano reversed phase LC (RPLC)-MS system with a large-i.d. trap setting was employed for proteomics analysis. The LC portion consists of a Dionex Ultimate 3000 nano LC system and an Ultimate 3000 gradient micro LC system with a WPS-3000 autosampler (Thermo Fisher Scientific, San Jose, CA, USA). Mobile phase A and B were 0.1% FA in 2% acetonitrile and 0.1% FA in 88% acetonitrile respectively. A volume of 4 μL samples were first loaded onto a large-i.d. trapping column (300 µm ID × 5 mm) with 1% B at 10 µL/min, and the trap was washed for 3 min. Samples trapped were flushed onto the nano LC column (75-µm ID × 100 cm) by a series of retrograde nanoflow gradients at 250 nL/min. The gradient profile used was: 4–13% B for 15 min; 13–28% B for 110 min; 28–44% B for 5 min; 44–60% B for 5 min; 60–97% B for 1 min and isocratic at 97% B for 17 min. To maximize chromatographic resolution and reproducibility, the nano LC column was heated to 52 °C. MS data acquisition was performed on a Thermo Orbitrap Fusion Lumos mass spectrometer (Thermo Fisher Scientific, San Jose, CA, USA) under data-dependent mode. MS1 spectra were acquired at 120 K resolution with an automated gain control (AGC) target of 500,000 and a max injection time of 50 ms. The most abundant precursors were selected for high-energy collision dissociation (HCD) at 35% normalized collision energy and Orbitrap MS2 detection. Precursors were filtered by quadrupole using an isolation window of 1 Th, and those interrogated in previous scan cycles were excluded using a dynamic window (60 s ± 10 ppm). MS2 spectra were acquired at 15 K resolution with an AGC target of 50,000 and a max injection time of 50 ms.

### 2.5. Protein Identification and Quantification

LC-MS rawfiles were searched against a combined Uniprot-SwissProt Mus musculus/Rattus norvegicus database (containing 24,991 protein entries) using MS-GF+ search engine (v10089, MS-GF+ search engine: Pacific Northwest National Laboratory, Richland, WA, USA, released on 16 July 2013). A target-decoy approach was employed for false positive rate (FDR) estimation and control. Primary searching parameters include: 20 ppm for precursor mass tolerance, −1 to 2 for isotope error range, fully-tryptic peptides only, 2 to 7 for precursor charge state, carbamidomethylation for fixed modification, the oxidation of methionine and acetylation of peptide N-terminal for variable modification. The other parameters follow the default setting of MS-GF+. Peptide filtering, global FDR control, and protein grouping was conducted by IDPicker (v3.1.643.0. https://proteowizard.sourceforge.io/idpicker/, accessed on 9 July 2015). The peptide number needed for protein identification was set to 1 for individual samples and 2 for merged results. Global protein-level FDR was set to 1%.

MS1 ion current-based quantification was accomplished by the IonStar data processing workflow featuring highly accurate quantification with low false-positives [26]. LC-MS rawfiles first underwent chromatographic alignment using the ChromAlign algorithm to correct inter-run retention time (RT) deviations. A direct ion-current extraction (DICE) method was employed to generate quantitative feature sets (i.e., frames) based on the *m*/*z* and RT in the aligned dataset. Both steps were performed in SIEVE^TM^ package (version 2.2 SP2, SIEVE: Thermo Fisher Scientific, San Jose, CA, USA). Frames with signal-to-noise ratio > 10 were retained for quantification and corresponding ion intensities were calculated. A customized R package, UHR-IonStar (https://github.com/JunQu-Lab/UHR-IonStar, accessed on 5 June 2021), was used to match quantitative features with identification results, perform dataset-wide normalization, reject outliers on the peptide level, and aggregate peptide-level data to protein level [27]. Protein ratio calculation and paired *t*-test were conducted manually in Microsoft Excel.

### 2.6. Bioinformatics Analysis

Bioinformatic analyses were performed using The Database for Annotation, Visualization and Integrated Discovery (DAVID) and Ingenuity Pathway Analysis (IPA) as described previously [25].

### 2.7. Western Blotting

Samples were lysed in radio immune precipitation assay (RIPA) buffer with protease inhibitor cocktail, PMSF and sodium orthovanadate (Santa Cruz Biotechnology, Santa Cruz, CA, USA). Twenty µg of proteins were loaded and resolved by SDS-PAGE and transferred onto nitrocellulose membrane. The membranes were blotted with primary antibodies, including anti-SERCA2 (1:1000, Cell Signaling Technology, Danvers, MA, USA), anti-FACL4 (1:1000, Abcam, Cambridge, UK), anti-Sigma-1R) (1:125, Abcam), and cytochrome c (1:1000, Cell Signaling Technology), overnight at 4 °C, followed by incubation with corresponding HRP-conjugated secondary antibodies. The membranes were developed with SuperSignal West Dura Chemiluminescence Substrate (Thermo Fisher Scientific Inc., Rockford, IL, USA) using ChemiDoc MP Imaging System (Bio-Rad, Hercules, CA, USA).

### 2.8. Statistical Data Analysis

Data are shown as mean ± SD. Statistical analysis was performed using two-tailed Student’s *t*-tests to compare two groups. Results of *p*-value less than 0.05 was considered significant.

## 3. Results

### 3.1. Isolation of MAM from Retinas of Non-Diabetic and STZ-Induced Type 1 Diabetic Rats

STZ-induced Long-Evans rats were used as an experimental model of Type 1 diabetes for studying retinal MAM in diabetes. This model was chosen because it has well-characterized pathological changes in the retina including retinal dysfunction, vascular leakage, and retinal degeneration [28,29,30], which recapitulate early human DR. Compared to the STZ-mouse model, STZ-diabetic rats also demonstrated more profound metabolic changes in the retina [31]. In addition, rat retinas are more suitable for MAM study given their much larger volume than mouse retinas. At 10 months after diabetes onset, the rats were used for retinal MAM proteomic study. The characterizations of blood glucose levels and body weights are shown in Table 1. MAM fractions were isolated from freshly harvested retinal tissue following a well-documented protocol [32,33] with minor modifications (Figure 1A). Western blot analysis showed that the MAM fractions are highly enriched with well-established MAM marker proteins including acyl-CoA synthetase 4 (FACL4) [34], Sigma-1R [35], and sarco/endoplastic reticulum Ca^2+^-ATPase (SERCA) [36]. A trace amount of cytochrome C was observed in the MAM [37], while the majority of the protein was localized to the purified mitochondria and in the pellet after MAM isolation (Figure 1B). These results attest to the high purity of the MAM sample.

### 3.2. Characterization of Altered MAM Proteins in STZ-Induced Diabetic Rat Retina

The analysis of MAM proteome has long been a daunting challenge largely due to the very limited sample amount and the high content-s of membrane proteins. To address this challenge, we employed IonStar, an integrative LC-MS proteomics approach, for MAM proteome profiling as described in our previous study [25]. IonStar encompasses three major components:(1)a surfactant cocktail-aided precipitation/on-pellet digestion (SEPOD) protocol for proteomics sample preparation;(2)a trapping nano LC-Orbitrap MS system for peptide separation and MS detection;(3)a data processing workflow to yield quantification results based on peptide MS1 ion currents.

The most prominent feature about IonStar is its excellent quantitative accuracy and reproducibility in large sample cohorts, which is resulted from the stringent controlled experimental procedures and well-optimized data processing workflow. LC-MS raw files were searched against a combined Uniprot-SwissProt Mus musculus/Rattus norvegicus database. In total, 2664 proteins were quantified with high confidence in the isolated MAM fractions of diabetic and non-diabetic rat retinas (Appendix A). The majority of these proteins were localized to either the cytoplasm (58.41%) or plasma membrane (22.97%) while a smaller minority were identified as nuclear (10.81%) or extracellular (3.53%) proteins. Among the total proteins identified, 179 proteins in STZ-induced diabetic rats exhibited significant changes in expression level (*p* < 0.05) as well as high biochemical relevance in pathophysiological states based on analyses using The Database for Annotation, Visualization and Integrated Discovery (DAVID) and Ingenuity Pathway Analysis (IPA). Interestingly, Gene Ontology (GO) analysis revealed that only 7% of the 179 altered MAM proteins were found in the cytosol. Rather, a large majority of the altered MAM proteins were localized to either the plasma membrane (37%), nucleus (18%), or mitochondria (10%), suggesting that plasma membrane-bound, nuclear, and mitochondrial proteins within the MAM complex are largely implicated in the disease process of DR (Figure 2A). Other less common sites of localization included the cytoskeleton (9%), endosome (6%), ER (5%), and Golgi apparatus (5%). A very small minority were associated with ribosomes (1%), peroxisomes (1%), and chromatin (1%).

Using DAVID bioinformatics and IPA, we further classified the 179 differentially expressed MAM proteins into their respective functional categories (Figure 2B). The majority of these proteins were identified as transporters (17.88%) or enzymes (17.88%). A number of specified enzymes—namely kinases (5.03%), phosphatases (2.23%), and peptidases (0.56%)—as well as ion channels (4.47%) and G-protein coupled receptors (3.91%), to name a few, comprised a smaller portion of the proteins. 41.34% of the altered MAM proteins had unidentifiable functions and were therefore categorized as “other.” The diversity of protein functions implicated in the altered MAM proteins in STZ-induced diabetic rat retina suggests an intricate network of signaling pathways and possible transcriptional, translational, and post-translational modifications that are involved in the pathogenesis of DR.

Further in-depth analysis using the above methods indicated that a large number of altered MAM proteins are instrumental in cellular processes, including but not limited to cell-cell signaling (31.84%), cell migration (29.61%), and cytoskeletal organization (25.70%) (Figure 3). Of note, only 1 of the 179 altered MAM proteins in diabetic rats was found to be involved in eliciting the ER stress response, demonstrating the relative lack of association between MAM processes in diabetes and consequent ER stress. This was an interesting finding in the context of our previous proteomic study in Type 2 diabetic mice, which highlighted ER stress as a potential cause of MAM proteome changes. In congruence with our findings in Figure 2B, many of the proteins were also involved in ion transport (22.91%), intracellular signal transduction (19.55%), and protein metabolism (25.14%), transport (16.76%), and synthesis (15.64%). Unsurprisingly, 29 proteins (16.20%) were associated with the inflammatory response—a key pathological mechanism underlying DR. Moreover, 16 proteins (8.94%) were associated with calcium regulation, which has strongly been implicated in neurodegenerative processes. Overall, it appears that these changes in protein levels may primarily orchestrate the processes of retinal inflammation and neuron death (Figure 4). As such, there is evidence that the MAM proteome may play a contributory role in the pathogenesis of Type 1 diabetes-induced retinal degeneration.

### 3.3. Pathological Processes Associated with Altered MAM Proteomes in STZ-Induced Diabetic Rat Retina

Given the numerous proteomic changes observed in the retinal MAM of STZ-induced diabetic rats, we focused on the top 25 proteins that exhibited the greatest changes in protein expression levels in the diabetic rat retina (Figure 5). A considerable portion of these proteins (32%) was tabulated to be involved in several of the hallmark pathological processes underlying DR, namely glucose metabolism dysfunction (CA14, HSPB1, GFAP, STEAP4), retinal degeneration (GPX4, RPE65, KCNJ13), neuroglial activation (LGALS3, GFAP), and fibrosis (LGALS3). In addition, we also identified any and all proteins among the 179 altered MAM proteins that are associated with a pathogenic process in DR, such as retinal ischemia and angiogenesis (Figure 5). Among all proteins that were significantly affected by the diabetic condition, most seem to be involved in either glucose metabolism or retinal degeneration, while a smaller number were found to play roles in fibrosis, neuroglial activation, retinal ischemia, and retinal angiogenesis. Several of the observed down-regulated proteins involved in retinal degeneration have previously been identified in various inherited retinal dystrophies, such as retinitis pigmentosa (RPE65, RP2, RBP3 and MERTK) [38,39,40], CRB1-retinal dystrophy [41], Usher syndrome (PCDH15), Leber congenital amaurosis (RD3) [42], and X-linked retinoschisis (RS1) [43], lending some support to the idea that there are a number of overlapping pathogenic mechanisms underlying both inherited retinal dystrophies and DR.

### 3.4. Bioinformatic Analyses of MAM Proteins Involved in DR

To visualize and identify functional interconnections among various MAM proteins involved in DR pathogenesis, we used IPA analysis to create protein-protein interaction maps that illustrated a vast interdependent network of MAM proteins isolated from the retina of STZ-induced Type 1 diabetic rats. In concordance with what we found regarding the MAM’s considerable association with the inflammatory response in diabetes, there were a significant number of interactions between altered MAM proteins in the diabetic condition and inflammatory mediators such as interleukin-1β (IL-1β), NF-kappa-B inhibitor alpha (NFκBIA), and Forkhead box protein O1 (FOXO1), which is known to regulate the response to oxidative stress [44] (Figure 6). Major associations were also discovered with endothelin 1 (EDN1), a potent vasoconstrictor, and c-Jun (JUN), a proto-oncogene.

Additionally, robust associations were identified between MAM proteins in DR and processes involved in cell homeostasis, glucose metabolism, and canonical DR-related pathological processes such as retinal neovascularization, retinal degeneration, and onset of hyperlipidemia (Figure 7). Other associated canonical DR-related processes included VEGF signaling, apoptosis, neuroinflammation, angiogenesis, and activation of reactive oxygen species (Figure 8). Most of the proteins within these maps were shown to be commonly affected by a handful of specific major proteins: SYVN1, STAT4, FOXO1, MAF, and LGALS3 (Figure 7 and Figure 8). Interestingly, reduced levels of SYVN1 have been reported to mediate DR pathogenesis via reduced inhibition of ER stress and vascular overgrowth [45]; STAT4 polymorphism has been shown to be associated with early-onset Type 1 diabetes [46]; FOXO1 is a key transcription factor known to induce insulin resistance in Type 2 diabetes [47]; MAF is highly expressed in pancreatic duct and islet cells [48]; and LGALS3 may exacerbate neuroinflammation in the diabetic retina and optic nerve in vitro [49]. It is notable that several of the altered MAM proteins are associated with genes that have been discovered to play roles in both Type 1 and Type 2 diabetes (e.g., STAT4 and FOXO1). When comparing the differentially expressed MAM proteins from our previous study in Type 2 diabetic mice with those of STZ-induced Type 1 diabetic rats, there were both remarkable differences in their MAM proteome profiles as well as several overlapping proteins between the two groups. As shown in Table 2, a distinct set of differentially expressed MAM proteins associated with retinal degeneration were found only in Type 1 diabetic rat retinas, among which are RP2, INSR, FGF2, RPE65, RBP3, RD3, GPX4, MERTK, PCDH15, CRB1, and KCNJ13. Some of these proteins are well-studied in retinal dystrophy, such as CRB1, RP2, RPE65, and PCDH15. Using Western blot analysis, we confirmed the presence of CRB1, RP2, and RPE65 in the MAM of rat retinas (Appendix A). Notably, two differentially expressed proteins GUCY2F and RS1 were found in both MAM fractions from Type 2 diabetic mouse brain or Type 1 diabetic rat retina, suggesting that these proteins may play a role in the pathogenesis of diabetes-associated neurodegeneration. Further investigation would be required to assess how various components of the MAM proteome may play diverse and/or similar roles in the pathology of DR and diabetic neurodegeneration in general in Type 1 and Type 2 diabetes.

The myriad of interconnections between the above major proteins and their respective associated MAM proteins in DR is likely parallel to a myriad of signaling pathways. Using IPA analysis, we further identified the specific signaling pathways that appear most implicated among the altered MAM proteins in STZ-induced diabetic rats (Table 3). We tabulated a total of 21 highly implicated signaling pathways in which as many as 14 to 39% of their associated protein mediators were down-regulated in the MAM of STZ-induced diabetic rats. A relatively smaller portion of protein mediators (1–11%) per signaling pathway was up-regulated in diabetic MAM. Several of the identified signaling pathways in our diabetic MAM samples are involved in general cellular maintenance processes, such as actin cytoskeleton signaling, integrin signaling, and RhoA signaling. However, most of the pathways can aptly be categorized as key processes underlying neuronal signaling and inflammation, cell proliferation, and systemic inflammatory responses.

## 4. Discussion

In our current study, we identified a total of 2664 proteins in the rat retinal MAM, among which 179 proteins exhibited markedly altered expression levels in STZ-induced Type 1 diabetes. Our study is the first to not only localize the differentially expressed MAM proteins in Type 1 diabetes but also extensively categorize, map, and detail their implications in the Type 1 diabetic condition using bioinformatic interrogation. We used DAVID bioinformatics and IPA analysis to determine associations between the altered MAM proteins in diabetes and important cellular processes, energy metabolism and generation, protein synthesis and trafficking, and calcium regulation. Further analysis of the changes in MAM protein expression illustrated the MAM’s dual roles in the pathogenesis of both disordered glucose metabolism and retinal degeneration. We delved deeper into the specifics of MAM protein-protein interactions and discovered key connections between many of the altered MAM proteins in diabetes with several major protein regulators of inflammation, diabetes, and/or DR, specifically IL-1β, NFκBIA, FOXO1, SYVN1, STAT4, MAF, and LGALS3. These findings have highlighted the MAM’s undeniable relevance in precipitating retinal complications known to occur in the Type 1 diabetic condition.

For the past few decades, studies have shown that DR is in large part a product of inflammation and retinal neural degeneration prior to microvascular complications that are found later in disease [2,3,4,5,6]. Numerous studies have zeroed in on the effects of single genetic aberrations in animal models of DR, all of which have lent greater insight into precisely how such inflammatory, neurodegenerative, and microvascular processes may unfold in DR [50]. By virtue of the advantages given to us by bioinformatic analyses, we were able to attain a more birds-eye view of the governing pathogeneses in DR as well as identify an abundance of fascinating correlations with our current knowledge of single-gene pathologies in diabetes and retinal degeneration. For example, some of the most significantly altered MAM proteins in our Type 1 diabetes model included LGALS3, GPX4, PDGFR, AQP4, and alphaA- and alphaB-crystallin (CRYAA and CRYAB). Specifically, we observed significant up-regulation in LGALS3, AQP4, and crystallin proteins and down-regulation in GPX4 and PDGFR. Interestingly, these changes in MAM protein levels paralleled current research that explores the implications of these genes in DR: increased levels of LGALS3 in Type 1 diabetes may exacerbate neuroinflammation in the retina and optic nerve as well as disrupt the blood-retinal barrier [49,51]; increased AQP4 has been associated with retinal edema in diabetic rats while decreased levels have neuroprotective effects in ischemia [52,53,54]; and retinal alpha- and beta-crystallins were highly expressed in human diabetic retina [55,56]. On the other hand, a reduction in PDGFR transcripts in mice recapitulated the classic features of non-proliferative DR, and high levels of GPX4 may have protective effects against high glucose conditions in DR, which suggests that decreased levels, as we observed in our proteomic data, may lead to the exacerbation of DR [57,58,59,60]. What is most interesting is that all such aberrations in protein levels were found simultaneously in our model of Type 1 diabetes. This suggests that DR is truly a large crossroad of multiple known pathologies. As such, our study may corroborate the idea that patients with Type 1 diabetes-associated DR may most benefit from therapies that take on an early and more generalized or multifactorial approach that targets the macro effects of hyperglycemia, as opposed to an approach that is mono-therapeutic and protein-specific.

Another fascinating finding in our proteomic study is the cross-association between alterations in the MAM proteome and a wide variety of inherited retinal dystrophies. As mentioned earlier, our proteomic data showed significant changes in the levels of proteins for retinitis pigmentosa (RPE65, RP2, MERTK, and RBP3), X-linked retinoschisis (RS1), CRB1-retinal dystrophy (CRB1), Usher syndrome (PCDH15), and Leber congenital amaurosis (RD3 and KCNJ13). Unlike in DR, which is an acquired retinopathy secondary to hyperglycemic conditions, inherited retinal dystrophies represent a category of diseases that stem from an inherited or de novo genetic mutation(s) that manifests a pathophysiology specific to that gene(s). What is interesting is that in both diseases, there appears to be evidence that acquired retinopathy due to diabetes and inherited retinal dystrophies may have overlaps in pathogeneses, given that many of the proteins that were down-regulated in diabetic MAM are those that are affected in inherited retinal dystrophies. Indeed, deeper investigation would be useful for determining how this combination of altered MAM proteins leads to the pathology that is seen in DR, as it is clear that DR does not clinically parallel each and every inherited retinal dystrophy whose associated genes were identified in this study. However, it is noteworthy to consider that, based on our findings, some similar pathophysiology may exist between DR and inherited retinal dystrophies.

One of the most significant ramifications of our study is the idea that the MAM may serve as a worthwhile vessel for investigating a host of DR-mediating proteins. One supportive example is our finding of elevated alpha-crystallin protein levels in the MAM in DR, which coincides with previous proteomic research performed on whole rat retina [61]. Most recently, studies in DR have shown that increased phosphorylation of CRYAA may be neuroprotective, in part by dramatically reducing ER stress as measured by reduced levels of eIF2α phosphorylation [62,63]. While our study did not specifically detect the post-translational modifications of MAM proteins, we were able to identify CRYAA as one of the differentially expressed proteins that comprises the MAM in Type 1 diabetic rat retina. Furthermore, we have confirmed the localization of CRYAA in the MAM by Western blot analysis (Appendix A). As perturbation of the MAM is well known to precipitate ER stress, our study, which localized CRYAA in the MAM, further supports the role of CRYAA in mediating the ER stress response. While such an example does not necessarily to suggest that the root of DR begins with aberrations in the MAM, it is to emphasize that by studying changes in the MAM in diabetes, which was warranted based on what we know of the MAM’s role in inflammation, we discovered a melting pot of previously suspected pathologies that were simultaneously implicated in DR.

Some limitations of our study include the use of limited MAM extracts due to the inherent difficulty in isolating large quantities of MAM fractions from rat retinal tissue. In addition, the retina is a high complex neural tissue composed by many distinct neuronal subtypes, each of which has unique protein profile and function. Our current study using MAM isolated from the whole retinal tissue would not be able to distinguish the origin of each of the proteins identified in the MAM proteome. Future studies would benefit from the inclusion of additional experiments to validate the proteomic changes in isolated neuronal subtypes. Alternatively, parallel studies using single cell RNA sequencing (scRNA-seq) or spatial transcriptomics can be carried out to correlate the MAM proteomic changes with cell type specificity of the retina. We may also employ greater numbers of rat retinal samples and/or supplement data with analogous rat brain samples from diabetic animals, as carried out in our previous MAM proteome experiment in Type 2 diabetic mice [25], to validate our current results. Intriguingly, several studies reported a low level of cytochrome C in the MAM and the ER from normal rodent tissues [37,64,65]. The underlying mechanisms are not clear but likely related to the multifaceted functions of this protein and its interaction with ER and MAM proteins such as inositol 1,4,5 triphosphate receptor (IP3R), a key component of the MAM tethering complex. This again supports a close interaction between the mitochondria and the ER via the MAM, the regulation and function of which warrants in-depth investigations.

In summary, our study is the first to perform a detailed proteomic profiling and analysis of retinal MAM in a rat model of Type 1 diabetes. Our quantitative analysis using bioinformatic interrogation suggested that diabetes, which is already a disease with one of the most multifaceted and numerous systemic complications in the human body, dysregulates a myriad of signaling pathways in the process of retinal degeneration alone. Future mechanistic studies of DR may benefit from examining proteomic changes in the MAM that parallel specific inflammatory and neurodegenerative processes that precipitate DR. Of investigative interest moving forward is to further examine how the MAM proteome interacts with the ER and mitochondria in both Type 1 and Type 2 diabetes.

## Figures and Tables

**Figure 1 cells-11-02819-f001:**
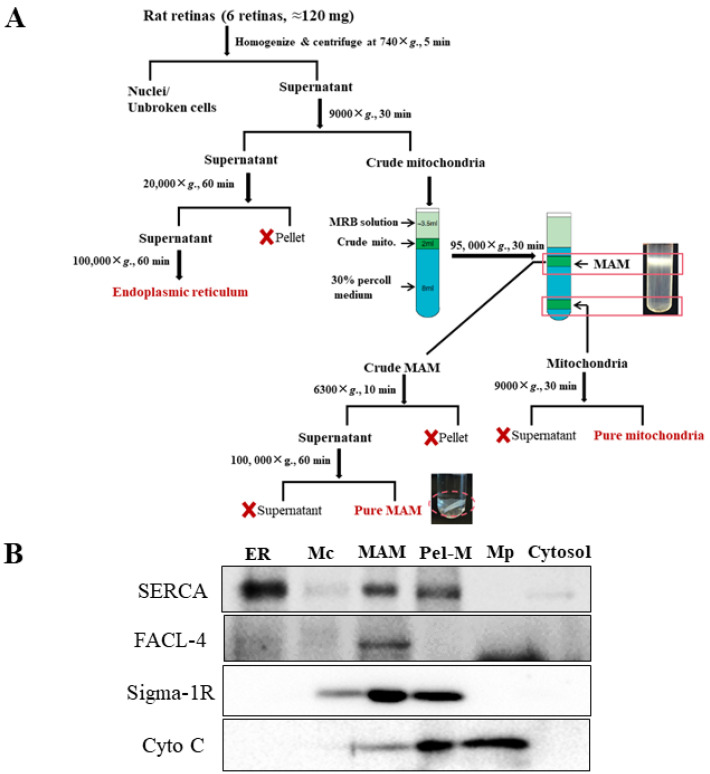
Schematic flowchart of the MAM isolation protocol and Western blot validation of MAM purity. (**A**) MAM was isolated from rat retina via homogenization, differential centrifugations, and a self-forming Percoll gradient centrifugation. ER, crude mitochondria, and pure mitochondria were also isolated following the multiple centrifugation steps. (**B**) Western blot analysis of MAM, ER and/or mitochondrial markers from retinal MAM fractions, ER, cytosol, and mitochondrial isolates. ER: endoplasmic reticulum, Mc: crude mitochondria, MAM: ER mitochondria-associated membrane, Pel-M: pellet after centrifugation of crude MAM, Mp: pure mitochondria, SERCA: Sarco/endoplasmic reticulum Ca^2+^-ATPase, FACL-4: Acyl-CoA synthetase 4, Sigma-1R: Sigma-1 receptor, Cyto C: cytochrome C.

**Figure 2 cells-11-02819-f002:**
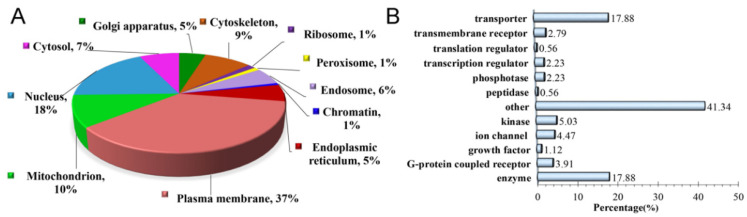
Subcellular localization and biological relevance of significantly altered MAM proteins in Type 1 diabetic rat retinas identified by proteomics. (**A**) Organelle association of the 179 altered MAM proteins based on cross-referenced annotation. (**B**) Percentages of molecule types, according to GO analysis from available database information. MAM: Mitochondria-associated ER membrane, GO: Gene ontology.

**Figure 3 cells-11-02819-f003:**
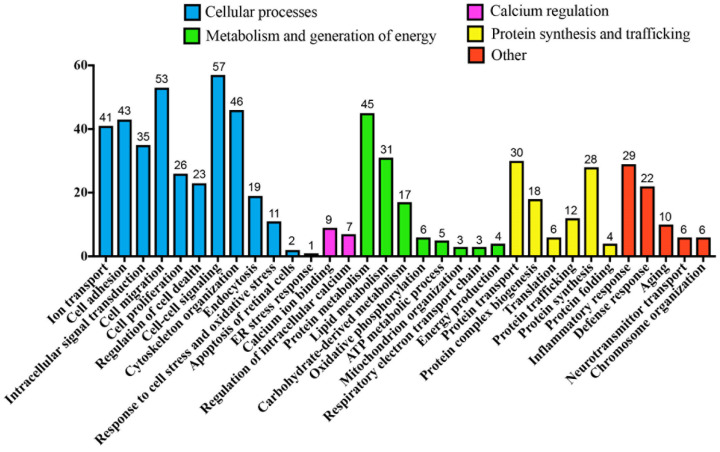
Hierarchical bar graph of key biological processes pertaining to significantly altered MAM proteins in Type 1 diabetic rat retinas. Biological processes of the 179 altered MAM proteins are classified as one of four categories of cellular activities or “other” according to GO analysis. MAM: Mitochondria-associated ER membrane, GO: Gene ontology.

**Figure 4 cells-11-02819-f004:**
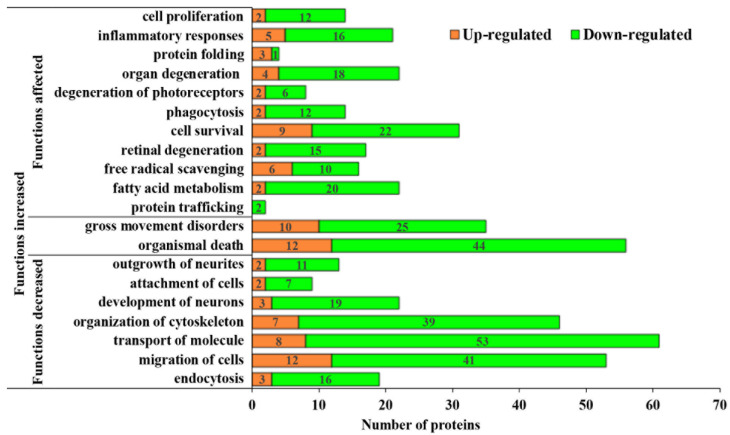
Bar graph of affected pathophysiologic processes that contribute to DR in the MAM of Type 1 diabetic rat retinas. MAM proteins are classified as up- or down-regulated as well as categorized into 20 biological processes that are labeled as functionally increased, decreased, or generally affected. DR: diabetic retinopathy, MAM: Mitochondria-associated ER membrane.

**Figure 5 cells-11-02819-f005:**
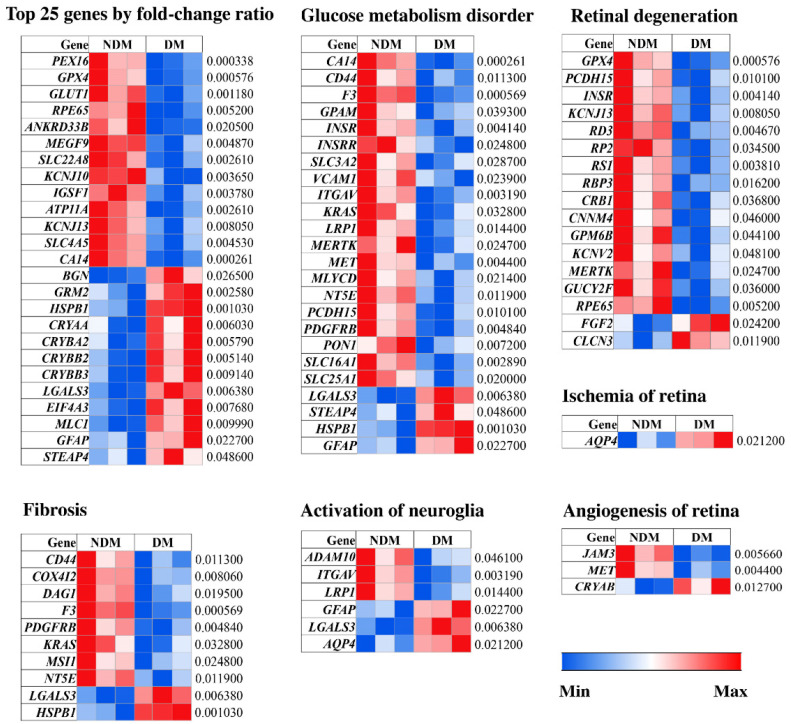
General and pathophysiology-specific heat maps of significantly altered MAM proteins in Type 1 diabetic rat retinas. MAM proteins that were most affected by fold-change ratio were selected among the 179 altered MAM proteins in the Type 1 diabetic condition. Proteins with a *p*-value of less than 0.05 were considered significant. *p*-values for each altered protein are shown on the right of each map. NDM: non-diabetes mellitus, DM: diabetes mellitus, MAM: Mitochondria-associated ER membrane.

**Figure 6 cells-11-02819-f006:**
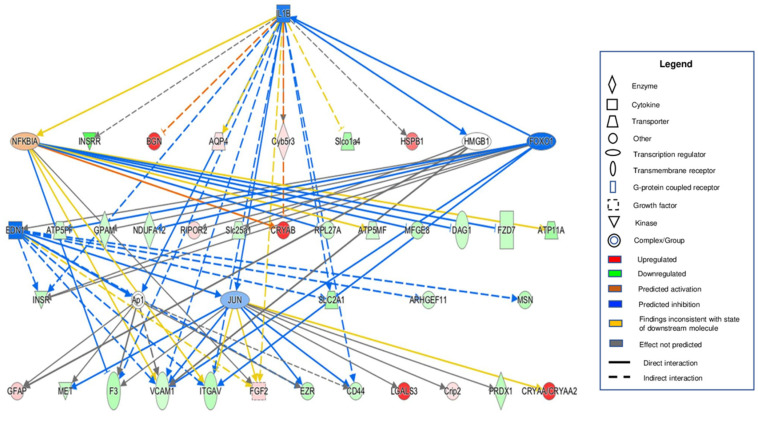
Interaction map depicting MAM proteins and their involvement with major inflammatory mediators. MAM proteins identified to have interactions with inflammatory mediators were selected from the 179 altered MAM proteins in the Type 1 diabetic condition. Numerous interactions were specifically identified with Interleukin-1β (IL-1β), NF-kappa-B inhibitor alpha (NF-κBIA), Forkhead box protein O1 (FOXO1), endothelin-1 (EDN1), and c-Jun (JUN). The functional networks were generated through IPA. Legend in the box displays molecules and function symbol, types, and colors. Lighter shades of red, green, and blue represent decreased intensities of up-regulation, down-regulation, and inhibition, respectively. MAM: Mitochondria-associated ER membrane, IPA: Ingenuity pathway analysis.

**Figure 7 cells-11-02819-f007:**
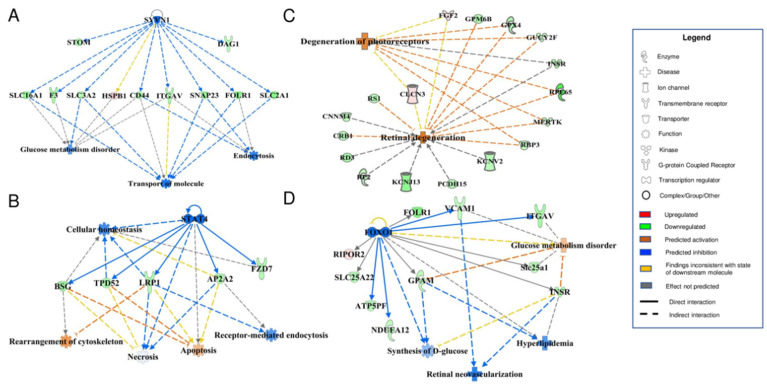
Interaction maps of MAM proteins and their multiple protein-protein associations with pathways of retinal degeneration. MAM proteins commonly identified to have protein-protein interactions with retinal degeneration processes were selected from the 179 altered MAM proteins in the Type 1 diabetic condition. (**A**) Eleven MAM proteins were found to have strong associations with SYVN1 as well as ties to glucose metabolism disorder and processes of molecular transport and endocytosis. (**B**) Five MAM proteins were found to have strong associations with STAT4 and multiple key intracellular processes. (**C**) Seventeen MAM proteins have common connections with processes of photoreceptor and retinal degeneration. (**D**) Ten MAM proteins each have associations with FOXO1 as well as intricate associations with glucose synthesis and metabolism, hyperlipidemia, and retinal neovascularization. The functional networks were generated through IPA. Legend displays molecules and function symbol types and colors. Lighter shades of red, green, and blue represent decreased intensities of up-regulation, down-regulation, and inhibition, respectively.

**Figure 8 cells-11-02819-f008:**
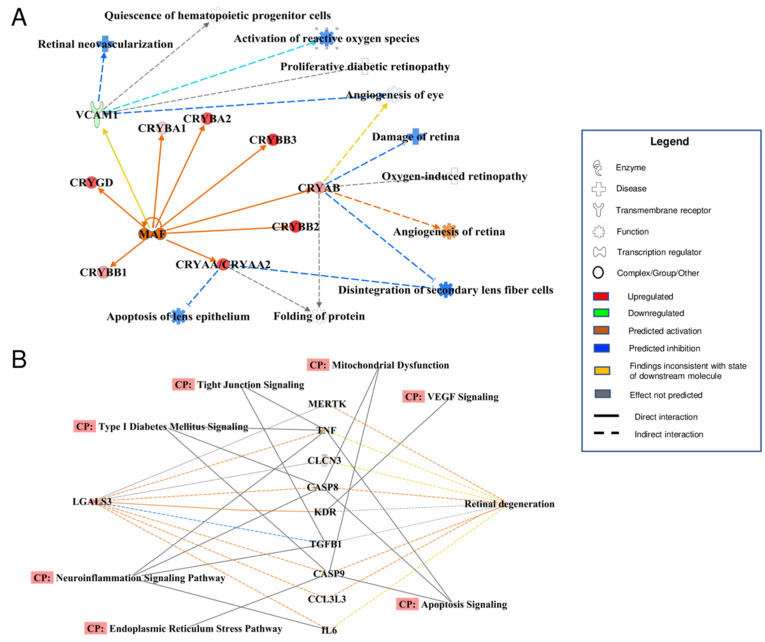
Interaction maps of MAM proteins and associated eye pathologies and canonical DR-related signaling pathways. MAM proteins with notable associations with specific signaling pathways and eye pathology underlying DR were selected from the 179 altered MAM proteins in the Type 1 diabetic condition. (**A**) Eight crystallin proteins and VCAM1 were notably identified as having associations with MAF. VCAM1 and several specific crystallin proteins (CRYAB, CRYAA/CRYAA2) were found to have numerous ties to the onset of various eye pathologies. (**B**) LGALS3 is a key MAM protein that is associated with processes of retinal degeneration via multiple specific protein-protein interactions. The functional networks were generated through IPA. Figure legend displays molecules and function symbol types and colors. Lighter shades of red, green, and blue represent decreased intensities of up-regulation, down-regulation, and inhibition, respectively. MAM: Mitochondria-associated ER membrane, IPA: Ingenuity pathway analysis, VCAM-1: Vascular cell adhesion molecule 1, MAF: MAF bZIP transcription factor, CRYAA: Crystallin alpha A, CRYAA: Crystallin alpha B, CRYAA2: Crystallin alpha A2, LGAL3: Galectin 3.

**Table 1 cells-11-02819-t001:** Body weight and blood glucose levels in diabetic and non-diabetic rats.

	NDM	DM	*p*-Value
*n*	9	9	
Body weight (g)	879.4 ± 47.6	411.3 ± 14.7	6.54 × 10^−8^
Blood glucose (mg/dL)	111.0 ± 4.2	377.3 ± 36.4	1.9 × 10^−6^

Data are shown as mean ± SE. NDM = non-diabetes mellitus; DM = diabetes mellitus.

**Table 2 cells-11-02819-t002:** Differentially expressed MAM proteins in retinal and brain tissues in animal models of Type 1 or Type 2 diabetes.

Biological Processes	Differentially Expressed MAM Proteins Observed Only in Mouse Brain with T2DM	Common Proteins	Differentially Expressed MAM Proteins Observed Only in Rat Retina with T1DM
Retinal degeneration	ABCA4, ATP1B2, GNAT1, GNGT1, GRK1, GUCY2D, PDE6B, PRPH2, RHO, SLC1A3, SLC6A6	GUCY2F, RS1	CLCN3, CNNM4, GPM6B, RP2, INSR, FGF2, RPE65, RD3, GPX4, MERTK, PCDH15, RBP3, KCNV2, CRB1, KCNJ13
Transport of amino acids	ARL6IP5, SLC17A7, SLC1A2, SLC1A3, SLC38A3, SLC6A9, SLC7A5, SLC32A1,	SLC3A2, KCNJ10, SLC7A8	SLC6A20, SLC7A10, INSR, SLCO1C1, FGF2, Slco1a4, SLC6A13, SLC25A22

**Table 3 cells-11-02819-t003:** Ingenuity Canonical Pathways associated with proteomic changes in retinal MAM with diabetes.

Ingenuity Canonical Pathways	−Log (*p*-Value)	Z-Score	Down-Regulated	Up-Regulated
Actin Cytoskeleton Signaling	1.94	−2.236	59/227 (26%)	17/227 (7%)
Integrin Signaling	1.45	−2.236	55/219 (25%)	18/219 (8%)
Leukocyte Extravasation Signaling	2.09	−2.236	40/211 (19%)	9/211 (4%)
EIF2 Signaling	1.99	−2	87/221 (39%)	9/221 (4%)
Glioblastoma Multiform Signaling	1.36	−2	33/160 (21%)	7/160 (4%)
Glioma Invasiveness Signaling	2.57	−2	17/70 (24%)	2/70 (3%)
IL-8 Signaling	2.23	−2	54/197 (27%)	9/197 (5%)
NF-κB Signaling	1.2	−2	25/181 (14%)	3/181 (2%)
RhoA Signaling	1.71	−2	40/124 (32%)	8/124 (6%)
STAT3 Pathway	2.49	−2	23/74 (31%)	3/74 (4%)
Gαi Signaling	5.16	−1.633	44/120 (37%)	8/120 (7%)
Signaling by Rho Family GTPases	2.34	−1.342	78/248 (31%)	17/248 (7%)
Synaptic Long Term Depression	2.13	−1.342	50/147 (34%)	10/147 (7%)
CREB Signaling in Neurons	2.36	−1	64/185 (35%)	14/185 (8%)
mTOR Signaling	2.19	−1	57/201 (28%)	11/201 (5%)
Synaptic Long Term Potentiation	1.75	−1	41/121 (34%)	13/121(11%)
ERK/MAPK Signaling	1.08	0	38/199 (19%)	9/199 (5%)
PPARα/RXRα Activation	1.21	1	46/180 (26%)	9/180 (5%)
PPAR Signaling	2.1	2	19/95 (20%)	1/95 (1%)
PTEN Signaling	1.77	2	27/119 (23%)	6/119 (5%)
RhoGDI Signaling21	3.23	2.236	61/173 (35%)	13/173 (8%)

## Data Availability

The mass spectrometry proteomics data have been deposited to the ProteomeXchange Consortium via the PRIDE partner repository with the dataset identifier PXD033040 and 10.6019/PXD033040. IonStar and the user manual are available at https://github.com/JunQu-Lab/UHRIonStarApp, accessed on 7 April 2022.

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
