# Peer review of "Proteomic Analysis of Retinal Mitochondria-Associated ER Membranes Identified Novel Proteins of Retinal Degeneration in Long-Term Diabetes"

_cells, 2022, doi:10.3390/cells11182819_

Round 1

Reviewer 1 Report

     This is an interesting, easy to read, well-made and potentially useful work. In it, a detailed and quantitative LC.MS-based proteomics profiling and analysis of differentially expressed retinal MAM proteins from a rat model of Type 1 diabetes is made, unveiling the subcellular localization of them, biological and pathophysiological process involved, heat and interaction maps of the involved proteins, among others. A careful bioinformatics analysis allowed authors to determine relationships between the altered MAM proteins in such pathology and important cellular processes from the energy metabolism, protein synthesis and trafficking and calcium regulation. Also, got evidences that diabetes dysregulates a large number of signalling pathways along the process of retinal degeneration, and that diabetic retinopathy is in large part a product of inflammation and retinal neurodegeneration prior to later microvascular complications. Several other useful conclusions related to the biochemical reasons behind this pathology and potential therapeutic approaches have also been reached. As said, a valuable work overall.

-This reviewer, in order to facilitate understanding by non-specialized readers, would suggest that authors explain a bit about the strategy behind the IonStar proteomics approach used here, which seems to be previously developed and published by some of the authors.

-Also, check whether the found percentages of the molecular types of MAM proteins, displayed in Fig.2, is correct for transporter and enzyme types, 17.88% in both cases, as well as for transcriptor regulator and phosphatase, 2.23% for each. Might such coincidences just be typing mistakes?  The word "phosphatose"  (phosphatase?) at the axis should also be checked.

-Given the numerous acronyms and abbreviations found in the work, sometimes not sufficiently explained like PERK and Tris-FA (found at lines 68 and 136), the collection of them in an Abbreviations Section should be considered.

-References 12, 19, 50, 53, 62 and 63 would require revision, mostly to complete volume and/or pages numbers.

Reviewer 2 Report

In this study, the authors have tried to identify retinal disease mechanisms with their basis in the mitochondria-associated endoplasmic reticulum (ER) membrane (MAM), that are associated with diabetic retinopathy. The authors used the exact same methods and computational analysis as they have used before (Reference 15, Sci. Rep. 2017) to purify mitochondria-associated ER membranes, categorize their protein content, and compare a diabetic rodent to a non-diabetic control. The only difference is that, in the current manuscript, they have purified it from retinal isolates (rather than brain), and they have induced type 1 diabetes in rats instead of using the Lepr db/db mouse model as they did in Ref. 15. Disappointingly, although the computational analysis seems sound, and the manuscript is well written, the entire study still remains highly descriptive, basically generating a resource for other studies/future research, without pinpointing the new discoveries in this study compared to the current knowledge of diabetic retinopathy. Several important controls are lacking, as indicated below, that are important in validating the approach as well as specifying the results.

Major comments:

(1) Most importantly, the authors should provide validations of a subset of their findings: they do not discuss the possibility that their preparations are not entirely pure, which would significantly reduce the relevance of all of the downstream computational analyses. They indicate they detect “a trace amount” of cytochrome-C in the MAM preparation, but it is a clear signal on western blot (Fig. 1B), indicating lots of other pollutants may also be detected as there is no cut-off for quantity. Therefore, it is of critical importance that they validate in the same way a significant subset of their identified MAM proteome members by detecting them in the different fractions on western blots.

(2) Importantly, they should include fractions from both the diabetic rat retinas as well as from the control rats, as the retinal changes caused by the diabetic retinopathy could affect the MAM isolation, which could skew the datasets.

(3) It is critical to include in the introduction, approach and discussion that “the retina” is not a single entity: it is a complex mixture of diverse cell types, organized in layers, all with unique functions. By just categorizing these processes for the entire retina, rather than taking the specific cell types into account, reduces the relevance and renders just long lists of altered processes without mechanistic relevance. Therefore, also markers of the different cell types should be taken along in the western blots, in order to get some idea if the used protocol is targeting the different cell types in the same manner.

(4) By identifying 2.664 unique proteins, all proposed to be active at the contact site between the ER and the mitochondria, they should provide some kind of rationale of why such a huge number would be required for the function. Again, in what way could cell type specificity contribute to this rationale? Of course, ideally, this study would be complemented by single cell RNAseq data or even spatial transcriptomics, as this is currently available, but even such studies that are published could be used to differentiate the current dataset for cell type specificity.

(5) Similarly, the many described pathways and protein networks are of limited value when their cellular context is not included: all that remains is the sense of complexity. In this light, it is important to more extensively discuss the comparison to their previous study (Ref. 15), and what the implications are of the differences shown in Table 2.

(6) Many of the identified proteins have not been described to localize to the MAM. The authors should provide validation for this for some of the most surprising candidates, such as the extracellular proteins, the nuclear proteins, and well-studied retinal dystrophy-associated proteins (e.g. CRB1, RP2, RPE65, PCDH15) which have never been demonstrated to localize to the MAM. Such an immunohistochemical validation/evaluation is critical to the validity of this study, and the value thereof to the research community.

(7) Access to the full list of all proteins, with their ID's and the measured MS values that form the basis of the figures and analyses, should be provided, e.g. as Excel tables. 

Minor comments:

(1) In the text (line 313), the authors mention the RP-associated protein RP3. Do they mean RP2 (listed in Figure 5) or RPGR (mutated in RP3)?

(2) Fig. 1: It is important to include an explanation of all abbreviations, including those of the antibodies, and what they mark/represent. These are now indicated in the text, but should also be included in the figure legend.

Round 2

Reviewer 2 Report

The revised manuscript still mainly repeats an earlier published proteomics approach (Reference 15, Sci. Rep. 2017) to determine the proteins present in the mitochondria-associated endoplasmic reticulum (ER) membrane (MAM), this time with from the retinal of a rat model for long-term diabetes type 1 vs, in the previous time, from the brain in a mouse model for long-term type 2 diabetes.

Most importantly, the authors have not included the data on several important controls I have proposed, except for a western blot analysis of four proteins out of the extensive list of proteins not known to localize to the MAM, but the western blots were from the same fractions used in the proteomics experiments, hence this is not a good control for (im)purity.

Without the proposed validations/controls/rationale, the bioinformatic analysis lacks foundation and conclusions remain not fully warranted.

Specific comments to the authors’ point-by-point responses:

(1) Please explain in the text that detection of a lower level of cytochrome C in the MAM is in line with previous findings (with the reference indicated in the response), as the signal shown in Figure 1 is very clear and to my opinion would not qualify as a “trace amount”. I do acknowledge the strive for a highly pure MAM fraction, and the markers shown in Figure 4 do show that the MAM fraction is significantly enriched for MAM-localized proteins, but the very high number 2.664 proteins in the MAM fraction does raise doubt on purity/specificity that needs to be controlled as proposed particularly with (2) and (6), which were not included.

(2) The authors seem to have misread my comment and therefore did not include the requested control: I did not propose to include an extensive proteomics study of other subcellular fractions, but the important visual comparison by western blot (of the controls for fractionation) shown in Fig. 1B, also for the diabetic rats (not just for the control rats as currently shown in the figure). I think this is a fair request for an important control that the authors should have performed as a validation of their samples before running the mass-spec.

(3) I agree to limit it to the proposed textual modifications here.

(4) I do not understand the response: the authors indicate that they have identified 2.664 unique proteins in the MAM, but in the next sentence of their response they indicate that they have NOT proposed that all these proteins are active at the contact site between the ER and the mitochondria. In the very first sentence of the abstract (lines 18-19), they indicate: “The mitochondria-associated endoplasmic reticulum (ER) membrane (MAM) is the physical contact site between the ER and the mitochondria”. Either the response is incorrect, or the indicated sentence. Assuming the former, I have to repeat my initial comment about the possible impurities. Moreover, there is no evidence that only proteins that show significant changes are MAM specific, hence the importance of validation of this specificity using a different technique, most commonly done by immunohistochemistry (as proposed in comment 6).

(5) As indicated in my introductory remarks, the western blots are from the same MAM fractions that were used for the proteomics experiments, hence it merely validates the MS-based detection of these proteins in these fractions, not that they are actually in the MAM (see: previous comment). Importantly, they again do not include the western blot comparison of even just the MAM fraction of the diabetic rat model, which would be an excellent way to validate the protein level differences measured by mass-spec.

(6) This is the most important and critical request for control/validation of specificity to the MAM, and it remains unclear why the authors indicate: “it is not possible to determine a protein localization to the MAM in the retina in vivo.“ All it takes is  the immunohistochemical staining of retinal section with antibodies specific for a few of the detected proteins, and show that they co-localize with MAM markers at the MAM. If this is not possible, the authors should at least indicate why not, as this is a very common procedure.

(7) Thank you for including this table: evaluating some of the ID’s without any cross-referencing within the table already show that proeins from all over the retinal do show up in the MAM fraction, e.g. rhodopsin and cone opsin, which are very likely pollutants, hence the request for validation.

Minor:

-          Golgi apparatus (line 258) is with a capital G.

-          Suppl. Fig. 1: Please use the same abbreviations as in Figure 1.

Author Response

August 31, 2022

Re: Resubmission of manuscript(R2). Cells] Manuscript ID: cells-1751401

Dear Editors,

Again, we would like to sincerely thank all reviewers, particularly reviewer 2 for spending time to re-evaluate our manuscript and provide further insightful comments. Please see below for our responses.

The revision has been developed in consultation with all coauthors, and each author has given approval to the final form of this revision. The agreement form signed by each author remains valid.

It is our belief that the manuscript is substantially improved after the revisions.

Thank you for your consideration.

Sincerely

Joshua J Wang, MD

  1. The revised manuscript still mainly repeats an earlier published proteomics approach (Reference 15, Sci. Rep. 2017) to determine the proteins present in the mitochondria-associated endoplasmic reticulum (ER) membrane (MAM), this time with from the retinal of a rat model for long-term diabetes type 1 vs, in the previous time, from the brain in a mouse model for long-term type 2 diabetes.

Response: In the original and revised manuscripts, we have discussed in great detail the significance of our study and the proteomic approach we used to examine MAM proteome. MAM is a critical subcellular compartment in regulation of calcium trafficking, oxidative stress, mitochondrial function and many other cellular processes. The protein composition of the retinal MAM has never been studied previously. Using IonStar, an integrative LC-MS proteomics approach with excellent quantitative accuracy and reproducibility, we determined the proteomic changes in rat retinal MAM from diabetic and non-diabetic animals. The results from this study provide valuable information and will serve as an important resource for researchers in the field of retinal research and diabetic retinopathy.

  1. Most importantly, the authors have not included the data on several important controls I have proposed, except for a western blot analysis of four proteins out of the extensive list of proteins not known to localize to the MAM, but the western blots were from the same fractions used in the proteomics experiments, hence this is not a good control for (im)purity.

Response: We provided specific response to the prior question regarding the purity. In addition, the western blot analysis was requested by Reviewer 2. We performed the experiment accordingly and provided the data in the rebuttal letter. We do not have remaining samples from the proteomics experiments. MAM and cell fraction samples were collected from additional animals. Please refer to prior rebuttal letter for further information.  

  1. Without the proposed validations/controls/rationale, the bioinformatic analysis lacks foundation and conclusions remain not fully warranted.

Response: We have provided validations for selected proteins using Western blot analysis. The rationale for the study was discussed extensively. Please also see response to comment 1. Please note that our proteomic study used retinal MAM samples isolated from 12-month diabetic rats. Age-/sex- matched non-diabetic rats were used as controls, which is highly appropriate.

Specific comments to the authors’ point-by-point responses:

(1) Please explain in the text that detection of a lower level of cytochrome C in the MAM is in line with previous findings (with the reference indicated in the response), as the signal shown in Figure 1 is very clear and to my opinion would not qualify as a “trace amount”. I do acknowledge the strive for a highly pure MAM fraction, and the markers shown in Figure 4 do show that the MAM fraction is significantly enriched for MAM-localized proteins, but the very high number 2.664 proteins in the MAM fraction does raise doubt on purity/specificity that needs to be controlled as proposed particularly with (2) and (6), which were not included.

Response: We have added the discussion in the text (lines 523 – 529). Thank you!

(2) The authors seem to have misread my comment and therefore did not include the requested control: I did not propose to include an extensive proteomics study of other subcellular fractions, but the important visual comparison by western blot (of the controls for fractionation) shown in Fig. 1B, also for the diabetic rats (not just for the control rats as currently shown in the figure). I think this is a fair request for an important control that the authors should have performed as a validation of their samples before running the mass-spec.

Response: Thank you for the clarification! We agree it is a fair request. Unfortunately, we do not have any extra MAM samples from 12-month diabetic rats for western blot analysis. As stated in our earlier response, the isolation of MAM from rodent retina is very challenging and requires large amount of tissue. All of the MAM samples from diabetic rats were used for proteomic analysis.

(3) I agree to limit it to the proposed textual modifications here.

Response: Thank you!

 (4) I do not understand the response: the authors indicate that they have identified 2.664 unique proteins in the MAM, but in the next sentence of their response they indicate that they have NOT proposed that all these proteins are active at the contact site between the ER and the mitochondria. In the very first sentence of the abstract (lines 18-19), they indicate: “The mitochondria-associated endoplasmic reticulum (ER) membrane (MAM) is the physical contact site between the ER and the mitochondria”. Either the response is incorrect, or the indicated sentence. Assuming the former, I have to repeat my initial comment about the possible impurities. Moreover, there is no evidence that only proteins that show significant changes are MAM specific, hence the importance of validation of this specificity using a different technique, most commonly done by immunohistochemistry (as proposed in comment 6).

Response: As discussed earlier, the LC-MS proteomics approach we used in our study, namely IonStar, has excellent sensitivity, quantitative accuracy and reproducibility (information provided in the text as requested by Reviewer 1). We respectively disagree that the large number of proteins are equivalent to possible impurities. For the latter, please see response to comment (1).  

(5) As indicated in my introductory remarks, the western blots are from the same MAM fractions that were used for the proteomics experiments, hence it merely validates the MS-based detection of these proteins in these fractions, not that they are actually in the MAM (see: previous comment). Importantly, they again do not include the western blot comparison of even just the MAM fraction of the diabetic rat model, which would be an excellent way to validate the protein level differences measured by mass-spec.

Response: Thank you for the comment. As discussed earlier, we do not have remaining MAM samples from 12-month diabetic rats for western blot analysis. In addition, we do not agree with the reviewer that western blot comparison would be an excellent way to validate the protein level differences measured by mass-spec. Compare to western blot analysis, proteomic approach, in particular with the cutting-edge systems such as IonStar, has substantially higher sensitivity and quantitative accuracy in quantification of protein changes. Western blot analysis can only provide semi-quantitative results and heavily rely on the availability of rigorously validated antibodies. For this reason, the results for many low-level proteins from western blot analysis often vary and are not reliable.

(6) This is the most important and critical request for control/validation of specificity to the MAM, and it remains unclear why the authors indicate: “it is not possible to determine a protein localization to the MAM in the retina in vivo.“ All it takes is the immunohistochemical staining of retinal section with antibodies specific for a few of the detected proteins, and show that they co-localize with MAM markers at the MAM. If this is not possible, the authors should at least indicate why not, as this is a very common procedure.

Response: Thank you again for the comment. Like western blot analysis, the immunohistochemical staining rely heavily on the specificity and sensitivity of the antibody. We have tested several antibodies for MAM markers in retinal IHC, but we have not found a good antibody for this purpose. Furthermore, the quantification of the detected proteins co-localized with the MAM markers adds another layer of challenge. We are working on optimizing this approach. To this end, we have some promising data (see below, for response only) in cultured R28 cells, a neuroprogenitors cell line. Our data show that RP2 is co-localized with Mfn2 (a MAM marker) and that treatment with high glucose for 7 days, but not for 48 hours, significantly reduced the level of RP2 in MAM. This result supports the finding in our proteomic study for a decreased RP2 in retinal MAM from 12-month diabetic rats. We plan to validate this result in a new in vivo study using diabetic animals and to investigate the role of RP2 in diabetic retinopathy.

(7) Thank you for including this table: evaluating some of the ID’s without any cross-referencing within the table already show that proeins from all over the retinal do show up in the MAM fraction, e.g. rhodopsin and cone opsin, which are very likely pollutants, hence the request for validation.

Response: Thank you for the comment. We would be happy to investigate the subcellular localizations of rhodopsin and cone opsin in our future studies. However, we do not agree that these proteins are likely pollutants just because their presence in the MAM fraction has not been reported previously or because the proteins from all over the retina show up in the MAM fraction. Again, the MAM was isolated from the whole retinal tissue. It is therefore not surprising to see the presence of retinal proteins from diverse cell types. Furthermore, photoreceptors are the most abundant cell population in the retina. We fully agree that there is a need to validate the localization of the proteins identified by the proteomic approach, but there are always unknown unknowns that need to be explored.  

Minor:

-          Golgi apparatus (line 258) is with a capital G.

Response: We corrected it with a capital G. Thank you! 

-          Suppl. Fig. 1: Please use the same abbreviations as in Figure 1

Response: We revised it to Supplemental Figure 1. Thank you!